# Ecologically Modified Leather of Bacterial Origin

**DOI:** 10.3390/ma17112783

**Published:** 2024-06-06

**Authors:** Dawid Lisowski, Stanisław Bielecki, Stefan Cichosz, Anna Masek

**Affiliations:** 1Institute of Polymer and Dye Technology, Faculty of Chemistry, Lodz University of Technology, 90-537 Lodz, Poland; 231401@edu.p.lodz.pl (D.L.);; 2International Center for Research on Innovative Biobased Materials, Lodz University of Technology, 2/22 Stefanowskiego Str., 90-537 Lodz, Poland; stanislaw.bielecki@p.lodz.pl

**Keywords:** bacterial nanocellulose, modification, rapeseed oil, grapeseed oil, linseed oil, plasticizing, additives

## Abstract

The research presented here is an attempt to develop an innovative and environmentally friendly material based on bacterial nanocellulose (BNC), which will be able to replace both animal skins and synthetic polymer products. Bacterial nanocellulose becomes stiff and brittle when dried, so attempts have been made to plasticise this material so that BNC can be used in industry. The research presented here focuses on the ecological modification of bacterial nanocellulose with vegetable oils such as rapeseed oil, linseed oil, and grape seed oil. The effect of compatibilisers of a natural origin on the plasticisation process of BNC, such as chlorophyll, curcumin, and L-glutamine, was also evaluated. BNC samples were modified with rapeseed, linseed, and grapeseed oils, as well as mixtures of each of these oils with the previously mentioned additives. The modification was carried out by passing the oil, or oil mixture, through the BNC using vacuum filtration, where the BNC acted as a filter. The following tests were performed to determine the effect of the modification on the BNC: FTIR spectroscopic analysis, contact angle measurements, and static mechanical analysis. As a result of the modification, the BNC was plasticised. Rapeseed oil proved to be the best for this purpose, with the help of which a material with good strength and elasticity was obtained.

## 1. Introduction

The increase in the manufacturing of leather fashion products from animal skin is gradually becoming a critical issue to sustainability for current and future generations [1]. Lately, Muthukrishnan [2] underlined the fact that despite the passage of years, leather still constitutes an important and valuable resource of national and international significance. This process might be easily evidenced with the data derived from the global statistical compendium for 1999–2015 by the Food and Agriculture Organization of the United Nations [3]. According to the document, nearly 1659.6 million bovine and 1163.7 million sheep from the overall livestock population were utilised by the leather industry. Apart from the edible parts, the hides and skins flayed accounted for nearly 6531.1 and 414.2 thousand tonnes. The hides were processed to 558.4 thousand tonnes of heavy leather and 14,298.7 million sq. ft of light leathers, relative to 5335.2 million sq. ft of leather processed from the skins. Between 2015 and 2020, there was an increase in the production of animal skins, with China being the world’s leading producer of skins [4]. 

Therefore, considering the fact that the emphasis is currently being put towards green technologies where the circular economy is driving innovation, the development of alternative sources for and the design of eco-friendly materials are essential [5,6,7]. Over the last decades, the gradual push toward the development of leather analogues has been trailed by scientific researchers and the footwear industry, leading to the design of various synthetic and natural materials [8]. Natural polymers are sought as alternatives because they meet the criteria of being both ecological and ethical [9]. Of all the natural polymers, plant-originated cellulose is known to be the most abundant on the planet [10,11]. However, aside from plants, cellulose can also be obtained from a variety of acetic-acid-producing bacterial strains belonging to, e.g., the genera Acetobacter xylinum [12], Gluconobacter oxydans [13], Gluconacetobacter xylinus [14], and Komagataeibacter medellinensis [15]. 

Compared to plant-derived cellulose, bacterial cellulose (BC) exists in a three-dimensional network [16]. Consequently, from a chemical point of view, this may provide an increased surface-area-to-volume ratio, hence, allowing for stronger interactions with surrounding components and moieties [17,18]. Additionally, from a physical point of view, the web-shaped sequence of monomeric units linked by β-1,4-glycosidic bonds might contribute to a high mechanical strength, an elevated degree of polymerisation, an increased crystallinity index (80–90%), tensile strength, and water-holding capacity compared to plant cellulose [19,20,21,22]. Therefore, BC fits not only biomedical solutions (e.g., for wound healing [23], antibacterial coatings [24,25], controlled drug delivery [26], cancer treatment [27], tissue engineering [28], and cell cultures [29]) but also finds some bulk applications [19], e.g., in the textile industries [30,31,32,33,34]. Consequently, selected technological solutions involving the use of bacterial cellulose in the textile and leather industries are presented below.

Firstly, the review carried out by Katyal et al. [30] showed that the use of BC in textile and footwear manufacturing is an innovative concept that avoids the use of animal skins and nonbiodegradable toxic materials. Based on various examples, BC exhibited the same elasticity and mechanical strength as animal skins commonly used in the footwear industry. Additionally, Kaminski et al. [31] described the use of glycerol to improve the flexibility of BC. The composites were prepared by immersion into a glycerol solution followed by high-temperature treatment to achieve a homogenous coating. Significantly, an increase in elongation at break and tensile strengths was demonstrated when comparing the treated samples to untreated samples. 

However, Phan et al. [32] went a one step further and showed leather-like BC-based materials synthesised by using a padding method that can confer three effects concurrently, which are colour, softness, and surface pattern. The leather-like BC/glycerol possessed remarkable colour strength (five times higher than that of other cellulosic textiles), sufficient porosity and softness (flexural rigidity of 35.15 ± 0.65 Nm; bending modulus of 135.54 ± 2.51 MPa), high tensile strength (26.46 ± 2.86 MPa), and relative resistance to water penetration at a level of approximately 0.09 ± 0.01 bar. 

On the other hand, Nguyen et al. [33] used slightly different approach to fabricate artificial leather. The authors of the study focused on the production of kombucha-derived bacterial cellulose (KBC) from different modified kombucha and diverse bio-waste sources using Komagataeibacter xylinus for high potential towards cost-effective industrial scalability. The prepared bio-based composites showed good shape stability and considerable flexibility, with an average tensile strength and elastic modulus of, respectively, 1.69 MPa and 100 MPa.

Moreover, Fernandes et al. [34] aimed at the development of BC-based composites containing emulsified acrylated epoxidised soybean oil (AESO) that were polymerised with a redox initiator system of hydrogen peroxide (H_2_O_2_) and L-ascorbic acid and ferrous sulphate as a catalyst. Importantly, the polymerisation of the emulsified organic droplets was tested before and after their incorporation into the BC by exhaustion. Furthermore, the obtained composites were characterised in terms of wettability, water vapor permeability (WVP), and mechanical, thermal, and antimicrobial properties. The authors remarked that when the AESO emulsion was polymerised prior to the exhaustion process, the obtained composites showed higher WVP, tensile strength, and thermal stability. On the other hand, the post-exhaustion polymerised AESO conferred the composite with a higher hydrophobicity and elongation.

The examples presented above demonstrate the successfulness of using bacterial cellulose in the leather industry. However, the methods used to produce BC-based artificial leather are still being developed [35,36]. The aim of the research in the article is to create a material that can replace animal skins. Therefore, the research presented here is an extension of previously known techniques and includes an analysis of the feasibility of using three natural oils from rapeseed, grapeseed, and linseed as plasticisers of bacterial cellulose. In addition, chlorophyll, curcumin, and an amino acid were also used as excipients. The substances chosen for the modification were selected based on their easy availability, natural origin, and low toxicity to humans. All the additives were incorporated into BNC using simple methods that did not require additional chemicals, making the material environmentally and people-friendly. Thanks to these properties, modified BNC could be a future replacement for leather not only in the fashion industry but also in prosthetics, for example, as a material for prostheses [37]. The artificial leather samples prepared were analysed for their mechanical properties, surface characteristics, and aesthetic considerations.

## 2. Materials and Methods

### 2.1. Materials

The subject of the study was bacterial bionanocellulose produced at the Institute of Molecular and Industrial Biotechnology, Lodz University of Technology. Oils of a vegetable origin were used for the modification, such as rapeseed oil produced by Komagara Sp. z o.o. (Warsaw, Poland), linseed oil obtained at Lidl sp. z o.o. (Tarnowo Podgórne, Poland), and grape seed oil distributed by Monini POLSKA Sp. z o.o. (Poznań, Poland). A 10% water solution of E141 chlorophyll produced by FOOD COLOURS (Piotrków Trybunalski, Poland) was used as an additive for the modification, and E100 curcumin in the form of turmeric oleoresin with an 8.00% curcumin content produced by FOOD COLOURS (Piotrków Trybunalski, Poland) was also used. Another of the additives used was L-glutamine with a purity of ≥98.5% supplied by Glentham Life Sciences (Corsham, UK). Toluene pure p.a produced by Chempur (Piekary Śląskie, Poland) was used as a solvent.

### 2.2. Sample Preparation

Samples of bacterial cellulose measuring 8 cm × 2 cm and approximately 0.5 cm thick were cut from sheets of bacterial bionanocellulose. For preservation and protection from the external environment, samples of pure unmodified bacterial bionanocellulose were stored in a 96% ethanol solution to prevent biodegradation of the samples. Before modification, the surface of the samples was dried of water and ethanol using a paper towel.

### 2.3. Modification Methods for Bacterial Bionanocellulose

#### 2.3.1. Filtration Method

A filter paper disc was placed in a Büchner funnel, which was placed in a Büchner flask. Samples prepared as described in Section 2.2 were placed on the filter paper. The whole sample was submerged in vegetable oil, and filtration was carried out. After the filtration, the samples were removed from the funnel and placed in a beaker with the oil, where a 24 h conditioning process was carried out. After this time, the samples were removed from the oil, gently dried with a paper towel, and placed in special frames designed to prevent the deformation of the BNC during the drying process. They were then placed in a laboratory dryer, where the drying process took place with a temperature of 60 °C and a drying time of about 24 h.

#### 2.3.2. Filtration Method Using Toluene

This is a modification of the filtration method using vegetable oil and toluene to modify the BNC. A filter paper disc was placed in a Büchner funnel, which was placed in a Büchner flask. Samples prepared as described in Section 2.2 were placed on the filter paper. The whole sample was flooded with vegetable oil and toluene mixed in a ratio of 1:1 (*v*/*v*), and filtration was carried out. After the filtration, the samples were removed from the funnel and placed in a beaker with the oil and toluene, where a 24 h conditioning process was carried out. After this time, the samples were removed from the oil, gently dried with a paper towel, and placed in special frames designed to prevent the deformation of the BNC during the drying process. They were then placed in a laboratory dryer, where the drying process took place with a temperature of 60 °C and a drying time of about 24 h.

### 2.4. Fourier Transform Infrared Spectroscopy (FT-IR) Absorbance Spectra Analysis

FT-IR spectroscopic analysis was performed to determine the absolute content of fatty acids in the sample. Fourier transform infrared spectroscopy (FT-IR) absorption spectra were obtained using a Thermo Scientific Nicolet 6700 FT-IR spectrometer with diamond Smart Orbit ATR sampling equipment in the range of 4000–400 cm^−1^. The number of used scans equaled 64 at a resolution of 4 cm^−1^. The absolute content of fatty acids in the sample was estimated from the FT-IR spectra using Equation (1), which allowed for the calculation of the carbonyl index. The carbonyl index is the quotient of the absorbance of the band derived from the carbonyl group (H_C=O_), which is located at a wavelength of about 1700 cm^−1^, and the absorbance of the band derived from the C−H group (H_C−H_), which is located at a wavelength of about 2800 cm^−1^.
(1)$$I=\frac{H_{C=O}}{H_{C-H}},$$

### 2.5. Contact Angle Measurements

The effect of the modification on the wettability of the BNC was investigated using an OCA 15EC goniometer from DataPhysics Instruments GmbH (Filderstadt, Germany). The instrument worked with an SCA 20 software module. Water was used as the measuring liquid during the measurement. A minimum of 5 similar CA results were obtained for each composite. The syringe chosen was a Braun DS-D 1000 SF equipped with a needle with an outer diameter of OD = 0.52 mm and an inner diameter of L = 0.25 mm and length of 38.10 mm. The volume of the liquid drops equaled 1 μL.

### 2.6. Static Mechanical Analysis

The effect of the modification on the mechanical strength of the samples was determined using a Zwick/Roell material testing machine. During the test, the speed of the moveable traverse of the machine was assumed to be stable and equal to 2 mm/min. Samples measuring 8 cm × 2 cm were placed in the jaws of the machine and then subjected to a ripping process. During this process, the software monitored the stress as a function of elongation, from which it determined the tensile strength and elongation at break of the samples. Three samples were tested for each modification. The samples tested were of non-normative dimensions, as some of the samples were destroyed when attempting to cut the shapes described in UNE EN ISO 527-1:2020-01 [38].

## 3. Results and Discussion

Two methods were used to modify the cellulose: the filtration method, where the amount of oil used for the modification was 150 mL, and a modification of the filtration method, in which vegetable oil and toluene were used to modify the bacterial nanocellulose (BNC) with amounts of 75 mL of vegetable oil and 75 mL of toluene. The modification methods and reagents used for the modification are shown in Table 1 and Table 2. The study was designed to determine the effect of rapeseed, linseed, and grape seed oils as well as the additives L-glutamine, chlorophyll, and curcumin on the properties of the BNC. The samples were subjected to tests such as FTIR spectroscopic analysis, contact angle measurements, and static mechanical analysis.

### 3.1. Fourier Transform Infrared Spectroscopy (FT-IR) Absorbance Spectra Analysis of Modified Bacterial Bionanocellulose Samples

#### 3.1.1. Samples of Bacterial Bionanocellulose Modified with Rapeseed Oil

The absolute content of fatty acids in the sample was determined by Fourier transform infrared spectroscopy. From the results, the carbonyl index of the samples was determined (Figure 1f). If the carbonyl index is high, it means that the sample contained carbonyl groups that originated from the modification oil, which means that the sample was saturated with oil. Each modification method, with the exception of modification with L-glutamine, increased the band height ~1700 cm^−1^, which was derived from the carbonyl group [39]. This means that each of the modification methods caused oil to penetrate the BNC structure. The other spectral bands visible in Figure 1 did not change and contained bands characteristic of cellulose [40].

Determining the saturation of the BNC with oil in the presence of additives was difficult because there were carbonyl groups in the additives used, which affected the intensity of the band at ~1700 cm^−1^. The spectrum (b) shown in Figure 1b was from the samples modified with rapeseed oil with L-glutamine and represents the spectrum characteristic of this additive [41]; this means that L-glutamine did not penetrate the structure of the BNC but was only deposited on the surface of the sample. The use of additives increased the carbonyl index. This was due to the presence of carbonyl groups in the chemical structure of the additives used. As the carbonyl index was used as an indicator of the presence of the oil used for modification in the structure of the modified BC, this did not affect the main objective of this work.

#### 3.1.2. Samples of Bacterial Bionanocellulose Modified with Linseed Oil

The BNC modified with linseed oil (Figure 2) had a low carbonyl index compared to the samples modified with rapeseed oil. This difference was due to their composition; these oils are composed of different fatty acids in different contents. In addition, the fatty acid content of the mixture can vary between oil batches and depends on factors influencing the breeding of the plants used to produce the oil. After the use of additives such as curcumin and chlorophyll, due to the presence of carbonyl groups in their structure, an increase in the carbonyl index was observed. As for the samples modified with rapeseed oil, in this case, L-glutamine also did not penetrate the BNC structure. The presence of toluene increased the intensity of the spectra of the linseed-oil-modified samples.

#### 3.1.3. Samples of Bacterial Bionanocellulose Modified with Grapeseed Oil

The BNC modified with grapeseed oil (Figure 3) showed a lower carbonyl index compared to the samples modified with rapeseed oil and, at the same time, a higher one compared to the samples modified with linseed oil. This difference could mean that the grape seed oil penetrated the BNC structure more but could also be due to the composition of the grape seed oil, which may have been influenced by the growing conditions of the plants from which the oil was produced and the way the oil was produced. The use of additives such as curcumin and chlorophyll increased the carbonyl index, which was due to the presence of carbonyl groups in the structure of these additives. As with the rapeseed- and linseed-oil-modified samples, L-glutamine did not penetrate the BNC structure.

In the FTIR spectra of the samples modified with rapeseed, linseed, and grape seed oils and additives such as chlorophyll, curcumin, and L-glutamate presented in Section 3.1, it can be seen that the additives chlorophyll and curcumin did not cause significant changes in the spectra. This was due to the chemical structure of these compounds; they contain the same chemical groups as cellulose and the oils used for the modification [25,42,43,44,45,46]. The only spectrum that stood out was that of the samples for which the aforementioned oils were used for the modification and L-glutamine, which did not penetrate the BC but was only deposited on the surface, as a result of which peaks corresponding to the chemical structure of L-glutamine were visible on the spectrum [41]. Excluding the spectra of the samples modified with L-glutamine, it can be seen that the analogous FT-IR spectra shown in Figure 1, Figure 2 and Figure 3 are very similar to each other. This is because all the oils used consist mainly of unsaturated fatty acids, meaning the molecules they contain have a similar chemical structure [43,46,47]. The influence of toluene on the spectra of the tested samples was also notable. Toluene caused an increase in the intensity of the spectra. However, this was not due to the presence of toluene in the samples, as the characteristic toluene band at 3030 cm^−1^ was not visible [48]. Due to its low surface energy and low density, the toluene caused a dilution of the mixtures used for the modification [49,50,51]. The dilution of the modification mixtures facilitated the penetration of the bacterial nanocellulose by the applied oils and additives, resulting in their higher concentrations in the samples and an increase in absorbance in the FT-IR spectra. An exception is the samples shown in Figure 1c, where the sample containing toluene exhibited a lower absorbance than the analogous sample without toluene. This may indicate the varying concentrations of the substances used for the modification in different parts of the sample.

### 3.2. Contact Angle Measurements of Modified Bacterial Bionanocellulose Samples

#### 3.2.1. Samples of Bacterial Bionanocellulose Modified with Rapeseed Oil

The effect of the rapeseed oil and additives on the wettability of the samples is shown in Figure 4. The samples with L-glutamine additives immediately absorbed the measurement liquid, as a result of which contact angle measurements were not possible. The presence of chlorophyll and curcumin additives increased the wettability of the material. The additives did not significantly affect the shape of the outer surface of the BNC, with the samples continuing to show ripples and defects that affected the wettability of the material.

#### 3.2.2. Samples of Bacterial Bionanocellulose Modified with Linseed Oil

The effect of the linseed oil and additives on the wettability of the samples is shown in Figure 5. Similar results to those observed for the samples modified with rapeseed oil were observed; the samples with L-glutamine immediately absorbed the measurement liquid, making it impossible to carry out contact angle measurements, while the presence of chlorophyll and curcumin additives increased the wettability of the material. A noticeable difference for the samples modified with rapeseed oil was the smaller measurement uncertainties for the samples with additives compared to the samples without additives. The additives caused the specimens to plasticise, resulting in no stresses, leading to microcracks and changes in the surface of the specimens, which occurred when the specimens shrank due to drying.

#### 3.2.3. Samples of Bacterial Bionanocellulose Modified with Grapeseed Oil

The effect of the grapeseed oil and additives on the wettability of the samples is shown in Figure 6. The effect of the grapeseed oil and additives was very similar to that of the linseed oil and additives. The presence of the chlorophyll and curcumin additives increased the repeatability of the measurements, as a result of which the measurement uncertainty decreased. This may mean that the surface layer of the samples was more homogeneous; there was less damage on it [52]. Modification with the addition of L-glutamine, as in the previously described cases, made contact angle measurements impossible.

The contact angles for each of the samples tested were below 90°, meaning that the samples exhibited hydrophilic properties. The presence of additives in each of the cases presented in Section 3.2 reduced the value of the contact angle relative to the samples modified with oil alone or oil with toluene without additives. The presence of toluene, which has a much lower surface tension than water, should theoretically increase the hydrophobicity of the samples by lowering the surface tension of water, which was the measurement liquid [51,53]. Such a tendency can be observed for the samples shown in Figure 4; for the other samples tested, the effect of toluene was not noticeable. This was due to the shape of the surface of the samples, which played a key role in the contact angle results [54,55]. The micro-cracks and depressions present on the samples’ surface significantly affected the contact angle results, which made it impossible to clearly determine what effect each of the substances used for the modification had on the contact angle results [52].

### 3.3. Static Mechanical Analysis of Samples of Modified Bacterial Bionanocellulose

#### 3.3.1. Samples of Bacterial Bionanocellulose Modified with Rapeseed Oil

The effect of the rapeseed oil and additives on the mechanical properties of the BNC was determined by static mechanical analysis tests (Figure 7). The addition of the chlorophyll had a positive effect on the mechanical properties of the sample compared to the reference sample, with an increase in the ultimate strength and elongation at break; the reference sample achieved an ultimate strength of 27.5 MPa and an elongation at break of 12.3%, while the sample modified with rapeseed oil and chlorophyll achieved an ultimate strength of 34.7 MPa and an elongation at break of 27%. The samples modified with rapeseed oil with added curcumin had a lower strength (16.9 MPa) and higher elongation at break (24.8%) compared to the reference samples. The presence of L-glutamine strengthened the samples and slightly increased their elongation at break compared to the reference sample; the strength enhancement may be due to the lower water content of the sample, and the L-glutamine did not penetrate the BNC but only covered its outer surface, which resulted in a significant expansion of the surface, and therefore the area from which the water could evaporate was larger, while the increase in the elongation at break may have been due to the plasticising properties of the rapeseed oil [56,57]. The presence of toluene adversely affected the properties of the samples, reduced their mechanical properties, and increased measurement uncertainties. The sample modified with rapeseed oil with the addition of chlorophyll had the lowest measurement uncertainties; this may indicate that the chlorophyll protected the sample from damage and the formation of microcracks and defects during the modification process.

#### 3.3.2. Samples of Bacterial Bionanocellulose Modified with Linseed Oil

The effect of the linseed oil and additives on the mechanical properties of the BNC was determined by static mechanical analysis tests (Figure 8). The modification with linseed oil and linseed oil with additives resulted in a significant decrease in breaking strength compared to the reference sample; nevertheless, the presence of the additives resulted in an increase in the elongation at break of the samples. The samples with chlorophyll and curcumin additives presented similar properties of breaking strength and elongation at break. Of the samples with additives, the samples containing L-glutamine had a lower elongation at break, while the breaking strength was similar compared to the samples modified with linseed oil with chlorophyll and curcumin. The presence of toluene in the samples with additives influenced a slight increase in the breaking strength of these samples.

#### 3.3.3. Samples of Bacterial Bionanocellulose Modified with Grapeseed Oil

The effect of the grape seed oil and additives on the mechanical properties of the BNC was determined by static mechanical analysis tests (Figure 9). The use of the grape seed oil modification resulted in a reduction in breaking strength relative to the reference sample. The presence of additives such as chlorophyll and curcumin had a positive effect on elongation at break. It can be seen that the modification with grapeseed oil with additives also reduced the measurement uncertainty of the tensile strength. The measurement uncertainty is the standard deviation of the fracture strength results of samples modified by the same method, which means that a large discrepancy in the strength results will result in a large uncertainty. The strength of the specimens made from the same material using the same method was most affected by microcracks and defects in the samples [58]. Therefore, the observed reduction in measurement uncertainty may mean that the additives used in the modification protected the specimens from damage and the formation of microcracks and defects during the modification process.

The figures in Section 3.3 show that the modifications with the additives of chlorophyll, curcumin, and rapeseed, linseed, and grape seed oils affected the samples in a similar way; these substances reduced the tensile strength and increased the elongation at the break of the samples. These changes were due to the fact that the BC was plasticised, i.e., plasticiser molecules penetrated between the BC chains, reducing the intermolecular forces and the energy required for molecular movement. Bacterial cellulose is made up of densely packed cellulose fibres that are intertwined with each other [59]. Once dried, the cellulose chains can form hydrogen bonds with each other, making the material stiffer and increasing its strength [60]. Plasticisation of a polymer involves the penetration of plasticiser molecules between the polymer chains, increasing the distance between chain segments and causing a decrease in the forces of intermolecular attraction. This phenomenon is illustrated in the diagram shown in Figure 10. Plasticisers can be substances of a medium or low molecular weight, such as the oils and additives used in this study [61]. The carboxylic acids present in the oils used for the modification penetrated between the cellulose chains, which caused the cellulose chains to separate and the material to plasticise. The additives used, due to the presence of -OH groups in their structure in the case of curcumin and −C=O in the case of chlorophyll, were capable of forming hydrogen bonds with the cellulose fibres, which prevented such bonds from forming between the cellulose fibres. Therefore, the effect of the oils and additives resulted in a decrease in breaking strength and a simultaneous increase in elongation at break. The use of the solvent (toluene) diluted the modification mixture and thus facilitated the penetration of plasticisers between the cellulose fibres, which, in the case of samples modified with rapeseed oil, led to excessive plasticisation and a large decrease in breaking strength. In contrast, the amino acid did not cause the plasticisation of the bacterial cellulose, as it did not penetrate between its chains but was only deposited on the surface of the samples tested.

## 4. Conclusions

The attempts to plasticise the BNC in this study were mostly successful, but this was nevertheless associated with a reduction in the strength of the samples relative to the reference sample, which was pure, unmodified dried BNC. The use of pure linseed and grape seed oils resulted in a deterioration of the strength and elasticity of the samples. When pure rapeseed oil was used, a decrease in the strength and an increase in the elasticity of the BC samples was observed. The use of each oil increased the hydrophobicity of the samples.

The additives chlorophyll, curcumin, and L-glutamine affected the properties of the BNC to a much greater extent than the use of oils alone. The additives affected the BNC samples similarly regardless of which oil they were used with. The addition of the amino acid mostly caused a decrease in strength and a decrease in elongation at break; this may have been due to the over-drying of the samples, which may have caused micro-cracking of the BC samples. The addition of curcumin was associated with a large decrease in the strength and a large increase in the elasticity of the BNC samples. The addition of chlorophyll increased the elasticity of the BNC samples and caused a decrease in strength, but these samples were the most durable of the oil-modified samples with additives. The additives had little effect on the wettability of the BNCs, excluding the amino acid, the presence of which prevented contact angle measurements because it did not penetrate the interior of the BNC but only remained on the sample surface where it absorbed the measurement liquid.

The research conducted in this study is an important step towards ecological and vegan design. Based on the results, it can be concluded that bacterial cellulose can be a substitute for animal skins in the future. Nevertheless, the modification process needs to be improved, and research needs to be conducted on the durability of the modified BNC. Further research will focus on improving the mechanical properties of the modified BNC and on improving the modification method so that it can be used in industry.

## Figures and Tables

**Figure 1 materials-17-02783-f001:**
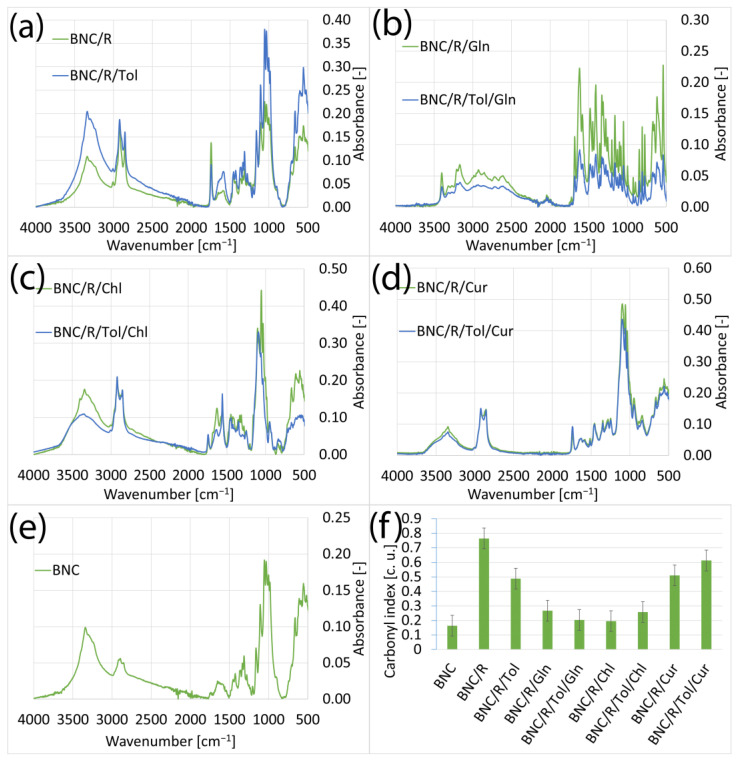
FTIR spectra of samples: (**a**) modified with rapeseed oil without additives (BNC/R), modified with rapeseed and toluene (BNC/R/Tol); (**b**) modified with rapeseed oil with added L-glutamine (BNC/R/Gln), modified with rapeseed oil with added L-glutamine and toluene (BNC/R/Tol/Gln); (**c**) modified with rapeseed oil with added chlorophyll (BNC/R/Chl), modified with rapeseed oil with added chlorophyll and toluene (BNC/R/Tol/Chl); (**d**) modified with rapeseed oil with added curcumin (BNC/R/Cur), modified with rapeseed oil with added curcumin and toluene (BNC/R/Tol/Cur); (**e**) unmodified bacterial cellulose; (**f**) carbonyl indexes of samples.

**Figure 2 materials-17-02783-f002:**
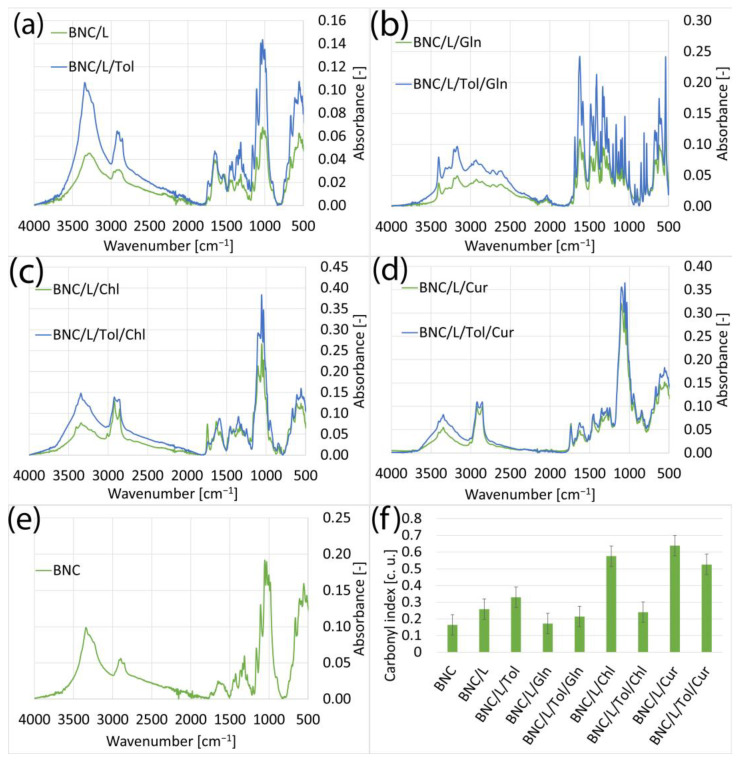
FTIR spectra of samples: (**a**) modified with linseed oil without additives (BNC/L), modified with linseed and toluene (BNC/L/Tol); (**b**) modified with linseed oil with added L-glutamine (BNC/L/Gln), modified with linseed oil with added L-glutamine and toluene (BNC/L/Tol/Gln); (**c**) modified with linseed oil with added chlorophyll (BNC/L/Chl), modified with linseed oil with added chlorophyll and toluene (BNC/L/Tol/Chl); (**d**) modified with linseed oil with added curcumin (BNC/L/Cur), modified with linseed oil with added curcumin and toluene (BNC/L/Tol/Cur); (**e**) unmodified bacterial cellulose; (**f**) carbonyl indexes of samples.

**Figure 3 materials-17-02783-f003:**
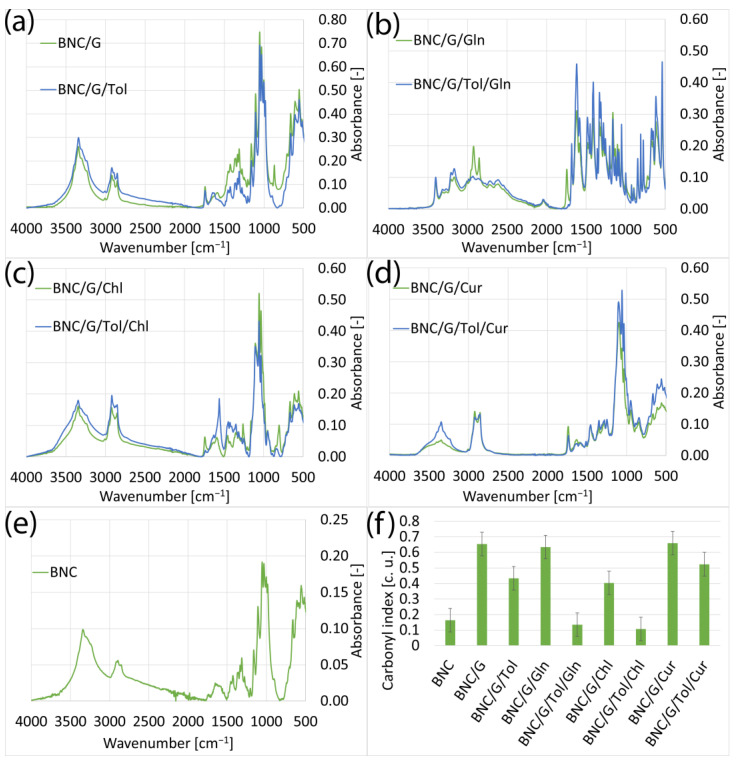
FTIR spectra of samples: (**a**) modified with grapeseed oil without additives (BNC/G), modified with grapeseed and toluene (BNC/G/Tol); (**b**) modified with grapeseed oil with added L-glutamine (BNC/G/Gln), modified with grapeseed oil with added L-glutamine and toluene (BNC/G/Tol/Gln); (**c**) modified with grapeseed oil with added chlorophyll (BNC/G/Chl), modified with grapeseed oil with added chlorophyll and toluene (BNC/G/Tol/Chl); (**d**) modified with grapeseed oil with added curcumin (BNC/G/Cur), modified with grapeseed oil with added curcumin and toluene (BNC/G/Tol/Cur); (**e**) unmodified bacterial cellulose; (**f**) carbonyl indexes of samples.

**Figure 4 materials-17-02783-f004:**
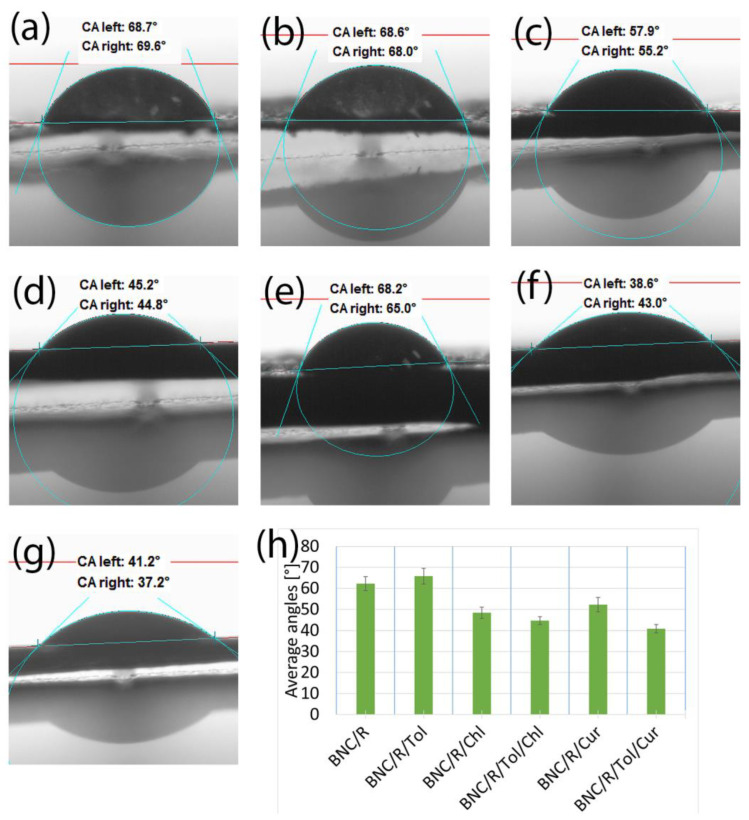
Contact angles of the samples: (**a**) modified with rapeseed oil without additives (BNC/R); (**b**) modified with rapeseed oil with toluene (BNC/R/Tol); (**c**) modified with rapeseed oil with added chlorophyll (BNC/R/Chl); (**d**) modified with rapeseed oil with added chlorophyll and toluene (BNC/R/Tol/Chl); (**e**) modified with rapeseed oil with added curcumin (BNC/R/Cur); (**f**) modified with rapeseed oil with added curcumin and toluene (BNC/R/Tol/Cur); (**g**) unmodified bacterial cellulose (BNC); (**h**) a compilation of the mean values of the contact angles for the samples listed.

**Figure 5 materials-17-02783-f005:**
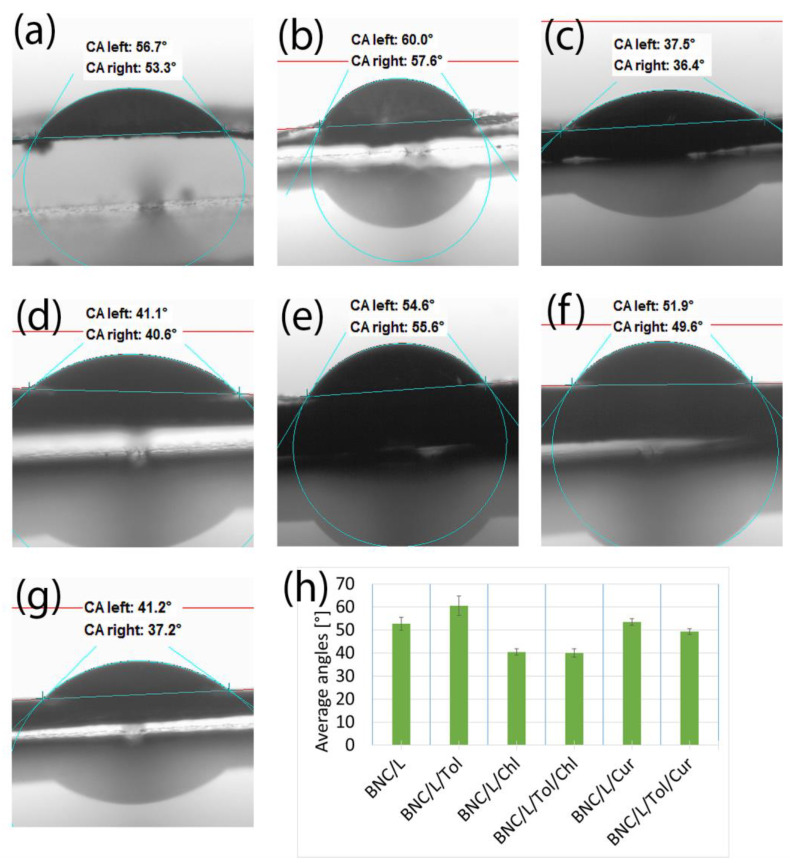
Contact angles of the samples: (**a**) modified with linseed oil without additives (BNC/L); (**b**) modified with linseed oil with toluene (BNC/L/Tol); (**c**) modified with linseed oil with added chlorophyll (BNC/L/Chl); (**d**) modified with linseed oil with added chlorophyll and toluene (BNC/L/Tol/Chl); (**e**) modified with linseed oil with added curcumin (BNC/L/Cur); (**f**) modified with linseed oil with added curcumin and toluene (BNC/L/Tol/Cur); (**g**) unmodified bacterial cellulose (BNC); (**h**) a compilation of the mean values of the contact angles for the samples listed.

**Figure 6 materials-17-02783-f006:**
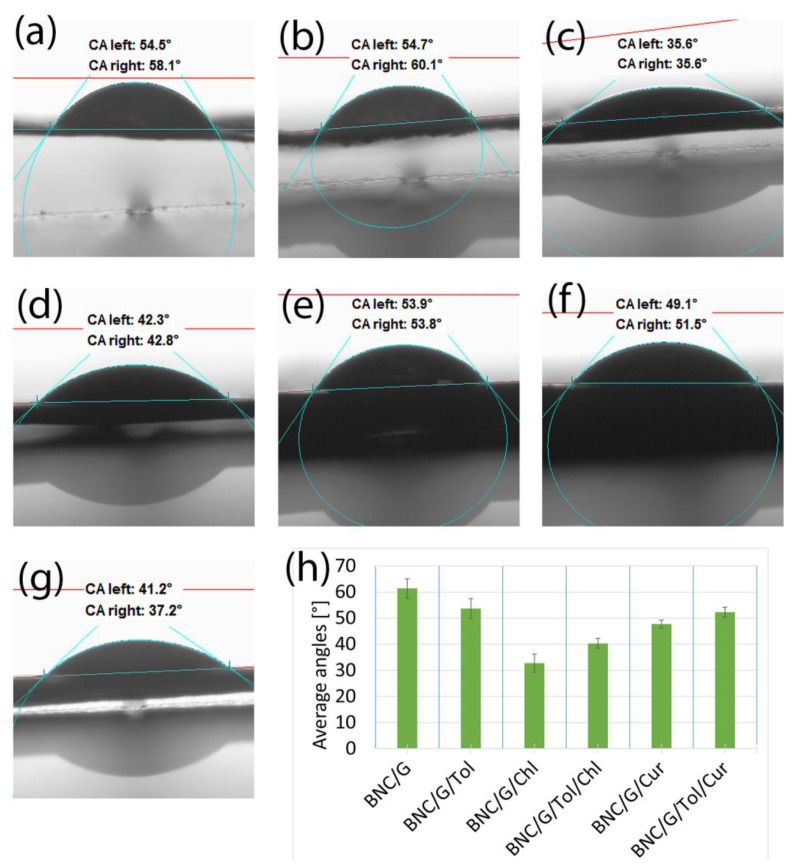
Contact angles of the samples: (**a**) modified with grapeseed oil without additives (BNC/G); (**b**) modified with grapeseed oil with toluene (BNC/G/Tol); (**c**) modified with grapeseed oil with added chlorophyll (BNC/G/Chl); (**d**) modified with grapeseed oil with added chlorophyll and toluene (BNC/G/Tol/Chl); (**e**) modified with grapeseed oil with added curcumin (BNC/G/Cur); (**f**) modified with grapeseed oil with added curcumin and toluene (BNC/G/Tol/Cur); (**g**) unmodified bacterial cellulose (BNC); (**h**) a compilation of the mean values of the contact angles for the samples listed.

**Figure 7 materials-17-02783-f007:**
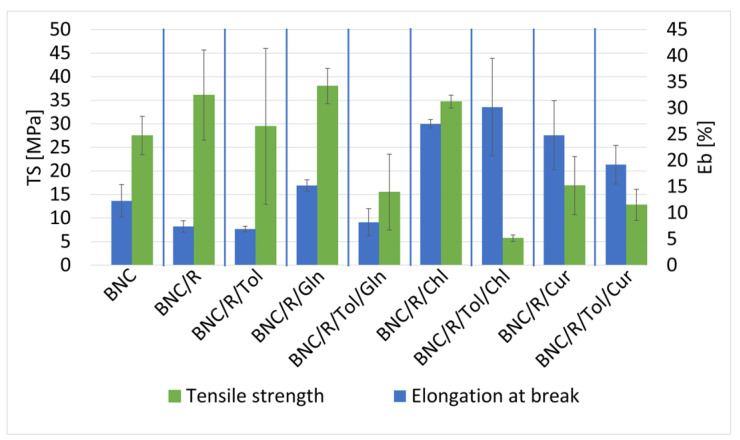
Tensile strength and elongation at break of samples: unmodified bacterial cellulose (BNC); modified with rapeseed oil without additives (BNC/R); modified with rapeseed and toluene (BNC/R/Tol); modified with rapeseed oil with added L-glutamine (BNC/R/Gln); modified with rapeseed oil with added L-glutamine and toluene (BNC/R/Tol/Gln); modified with rapeseed oil with added chlorophyll (BNC/R/Chl); modified with rapeseed oil with added chlorophyll and toluene (BNC/R/Tol/Chl); modified with rapeseed oil with added curcumin (BNC/R/Cur); modified with rapeseed oil with added curcumin and toluene (BNC/R/Tol/Cur).

**Figure 8 materials-17-02783-f008:**
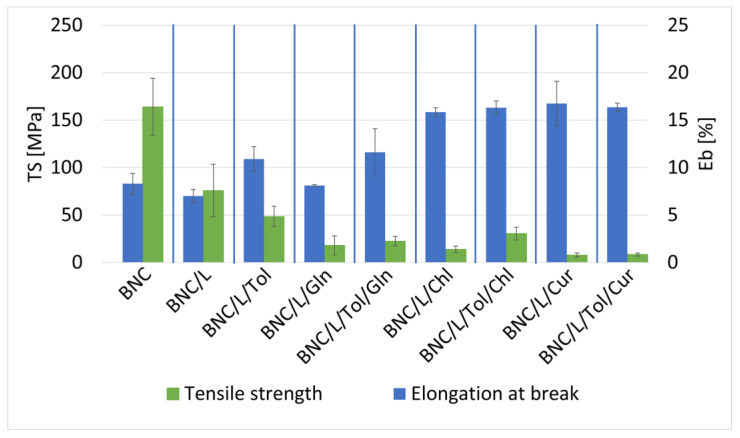
Tensile strength and elongation at break of samples: unmodified bacterial cellulose (BNC); modified with linseed oil without additives (BNC/L); modified with linseed and toluene (BNC/L/Tol); modified with linseed oil with added L-glutamine (BNC/L/Gln); modified with linseed oil with added L-glutamine and toluene (BNC/L/Tol/Gln); modified with linseed oil with added chlorophyll (BNC/L/Chl); modified with linseed oil with added chlorophyll and toluene (BNC/L/Tol/Chl); modified with linseed oil with added curcumin (BNC/L/Cur); modified with linseed oil with added curcumin and toluene (BNC/L/Tol/Cur).

**Figure 9 materials-17-02783-f009:**
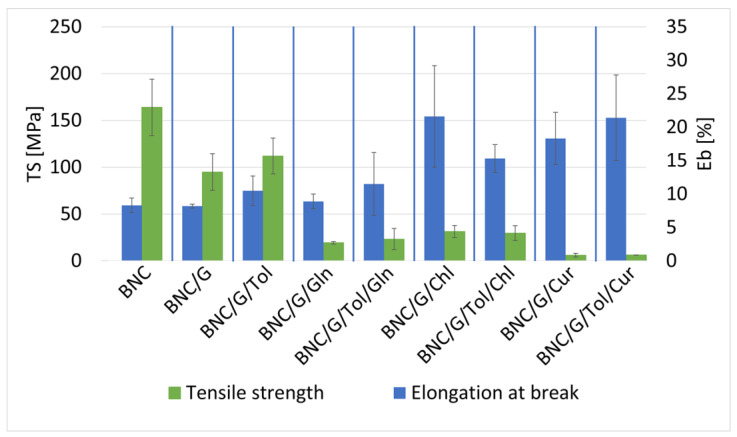
Tensile strength of samples: unmodified bacterial cellulose (BNC); modified with grapeseed oil without additives (BNC/G); modified with grapeseed and toluene (BNC/G/Tol); modified with grapeseed oil with added L-glutamine (BNC/G/Gln); modified with grapeseed oil with added L-glutamine and toluene (BNC/G/Tol/Gln); modified with grapeseed oil with added chlorophyll (BNC/G/Chl); modified with grapeseed oil with added chlorophyll and toluene (BNC/G/Tol/Chl); modified with grapeseed oil with added curcumin (BNC/G/Cur); modified with grapeseed oil with added curcumin and toluene (BNC/G/Tol/Cur).

**Figure 10 materials-17-02783-f010:**
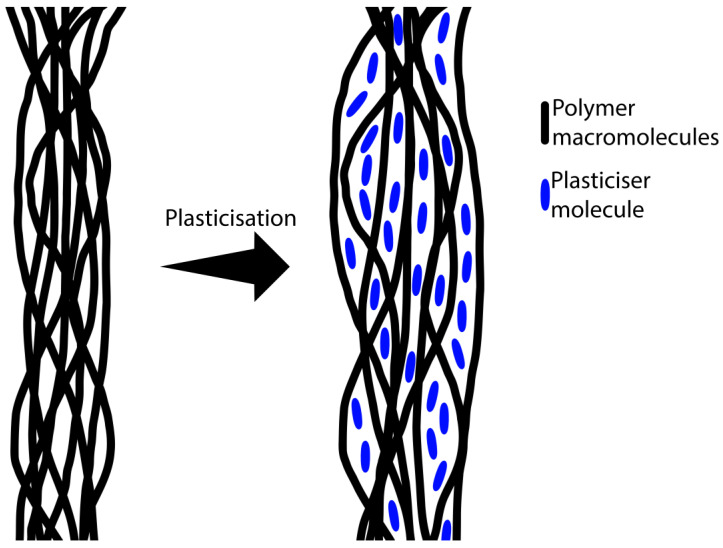
Diagram showing a simplified plasticisation mechanism.

**Table 1 materials-17-02783-t001:** Oils and mixtures used in filtration method.

Nº	Oil Used	Oil Volume [mL]	Additive Used	Amount of Allowance
1	Rapeseed	150	No	-
2	Linseed	150	No	-
3	Grape seeds	150	No	-
4	Rapeseed	150	Chlorophyll	5 mL
5	Rapeseed	150	Curcumin	5 mL
6	Rapeseed	150	L-glutamine	5 g
7	Linseed	150	Chlorophyll	5 mL
8	Linseed	150	Curcumin	5 mL
9	Linseed	150	L-glutamine	5 g
10	Grape seeds	150	Chlorophyll	5 mL
11	Grape seeds	150	Curcumin	5 mL
12	Grape seeds	150	L-glutamine	5 g

**Table 2 materials-17-02783-t002:** Oils and mixtures used in filtration method using toluene.

Nº	Oil Used	Oil Volume [mL]	Solvent	Amount of Solvent [mL]	Additive Used	Amount of Dditive
1	Rapeseed	75	Toluene	75	No	-
2	Rapeseed	75	Toluene	75	Chlorophyll	5 mL
3	Rapeseed	75	Toluene	75	Curcumin	5 mL
4	Rapeseed	75	Toluene	75	L-glutamine	5 g
5	Linseed	75	Toluene	75	No	-
6	Linseed	75	Toluene	75	Chlorophyll	5 mL
7	Linseed	75	Toluene	75	Curcumin	5 mL
8	Linseed	75	Toluene	75	L-glutamine	5 g
9	Grapeseed	75	Toluene	75	No	-
10	Grapeseed	75	Toluene	75	Chlorophyll	5 mL
11	Grapeseed	75	Toluene	75	Curcumin	5 mL
12	Grapeseed	75	Toluene	75	L-glutamine	5 g

## Data Availability

The original contributions presented in the study are included in the article, further inquiries can be directed to the corresponding author.

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
