# Peer review of "Ecologically Modified Leather of Bacterial Origin"

_materials, 2024, doi:10.3390/ma17112783_

Round 1
Reviewer 1 Report
Comments and Suggestions for Authors
Abstract
(Line 10) – The abstract should provide a concise summary of the entire article, including the Introduction, Objective, Material and Methods, Results and Discussion, and Conclusion. I recommend revising the objective to be more specific, presenting the methodology in a clear and logical sequence, highlighting the main results, and indicating the materials that performed the best. The discussion and conclusion should be based on the reported values.
Introduction
(Lines 24 to 29) – The introduction should provide a clear context for the research, explaining its significance and relevance. It's crucial to reference the information presented to support its validity. The current text seems more like a methodology than an introduction and does not contribute to a better understanding of the article. I suggest removing the paragraph.
(Line 89) - Replace “(H2O2)” with “(H2O2)”. Put the number 2 in subscript form.
At the end of the introduction, it's important to clearly state the research objectives. These should be direct, concise, and not include information about the steps developed in the Materials and Methods. The introduction should also highlight the novelty of the research and the potential benefits it can bring to the industry, environment, and society.
Material and methods
Did the sample preparation follow the determination of any specific literature or technical standard? If yes, add to work. Was equation 1 taken from any technical standard or literature? It is important to highlight the source in the form of a citation. Indicate the meaning of the unknowns by referring to equation (1) in the work text. How many samples were tested for each material evaluated in static mechanical analysis? How many repetitions were performed? What are the dimensions of the samples? Why did the authors not use technical standards to carry out the tests?
Results and discussion
(Line 171) – Remove the topic of word abbreviations. Words and abbreviations must be inserted into the text at the time of citation. Tables 1 and 2 – Replace "NO" with Nº.Why do the authors only discuss the results referring to Figure 1f? Isn't the other information important for the job? Improve the discussion of the results, highlighting the most important information in each of the graphs in Figure 1. I suggest that the authors search the literature for citations to contribute to the results and observations presented. Improve image quality to 300 dpi.(In topics: (3.1.2), (3.1.3)) - Improve the discussion of the results, highlighting the most important information in each of the graphs in Figure 2. I suggest that authors search the literature for citations to contribute to the results and observations presented. Improve image quality to 300 dpi. Better organize the graphics in Figures 4h, 5, and 6 h, and improve the quality of the images to 300 dpi. Adding more citations from other researchers that contributed to justifying the events, statements, and information presented was necessary. I suggest that authors search the literature for citations to contribute to the results and observations presented. Better organize the graphics in Figures 7, 8, and 9, and improve the quality of the images to 300 dpi. I suggest that authors search the literature for citations to contribute to the results and observations presented.
Conclusions
Add the research's positive points and the potential transformative benefits it can promote. Present some inspiring suggestions for future studies.
References
The article's references do not currently comply with the journal's standards. I suggest the authors to review the standards presented in the submission template and make the necessary corrections. This will ensure the credibility and robustness of your research, giving you and your work the confidence it deserves.
Author Response
PhD, DSc Anna Masek
Institute of Polymer and Dye Technology
Lodz University of Technology
90-537 Lodz, Stefanowskiego 16, Poland
Tel.: +48 42 631 32 23, Fax: +48 42 636 25 43
May 30, 2024
Dear Editor,
We resubmit our revised paper entitled “Ecologically modified leather of bacterial origin” bywith a request for reconsideration for publication in "Materials” journal.
We have carefully analyzed the comments of the reviewers. The manuscript was corrected exactly according to them. A list of responses to reviewers’ comments and corrections made in the manuscript are attached. In the manuscript, changes made based on the reviewers' comments are marked in red.
The manuscript has not been previously published, is not currently submitted for review to any other journal, and will not be submitted elsewhere before a decision is made by this journal.
For correspondence please use the following information:
Corresponding author: Anna Masek
Institute of Polymer and Dye Technology
Lodz University of Technology
90-537 Lodz, Stefanowskiego 16, Poland
Tel.: +48 42 631 32 93
Fax: +48 42 636 25 43
e-mail: anna.masek@p.lodz.pl
Yours sincerely,
PhD, DSc Anna Masek
Answers to reviewer’s comments
Reviewer #1
- Abstract
(Line 10) – The abstract should provide a concise summary of the entire article, including the Introduction, Objective, Material and Methods, Results and Discussion, and Conclusion. I recommend revising the objective to be more specific, presenting the methodology in a clear and logical sequence, highlighting the main results, and indicating the materials that performed the best. The discussion and conclusion should be based on the reported values.
Answer: We agree with the Reviewer’s comment. It has been improved.
- Introduction
(Lines 24 to 29) – The introduction should provide a clear context for the research, explaining its significance and relevance. It's crucial to reference the information presented to support its validity. The current text seems more like a methodology than an introduction and does not contribute to a better understanding of the article. I suggest removing the paragraph.
(Line 89) - Replace “(H2O2)” with “(H2O2)”. Put the number 2 in subscript form.
At the end of the introduction, it's important to clearly state the research objectives. These should be direct, concise, and not include information about the steps developed in the Materials and Methods. The introduction should also highlight the novelty of the research and the potential benefits it can bring to the industry, environment, and society.
Answer: We agree with the Reviewer’s comment. It has been improved.
- Material and methods
Did the sample preparation follow the determination of any specific literature or technical standard? If yes, add to work. Was equation 1 taken from any technical standard or literature? It is important to highlight the source in the form of a citation. Indicate the meaning of the unknowns by referring to equation (1) in the work text. How many samples were tested for each material evaluated in static mechanical analysis? How many repetitions were performed? What are the dimensions of the samples? Why did the authors not use technical standards to carry out the tests?
Answer: We agree with the Reviewer’s comment. It has been improved. Samples were not prepared to technical standards. Three samples of each material were tested in static mechanical analysis. The samples were 8cm x 3cm in size.
- Results and discussion
(Line 171) – Remove the topic of word abbreviations. Words and abbreviations must be inserted into the text at the time of citation. Tables 1 and 2 – Replace "NO" with Nº.Why do the authors only discuss the results referring to Figure 1f? Isn't the other information important for the job? Improve the discussion of the results, highlighting the most important information in each of the graphs in Figure 1. I suggest that the authors search the literature for citations to contribute to the results and observations presented. Improve image quality to 300 dpi.(In topics: (3.1.2), (3.1.3)) - Improve the discussion of the results, highlighting the most important information in each of the graphs in Figure 2. I suggest that authors search the literature for citations to contribute to the results and observations presented. Improve image quality to 300 dpi. Better organize the graphics in Figures 4h, 5, and 6 h, and improve the quality of the images to 300 dpi. Adding more citations from other researchers that contributed to justifying the events, statements, and information presented was necessary. I suggest that authors search the literature for citations to contribute to the results and observations presented. Better organize the graphics in Figures 7, 8, and 9, and improve the quality of the images to 300 dpi. I suggest that authors search the literature for citations to contribute to the results and observations presented.
Answer: We agree with the Reviewer’s comment. All graphics in the manuscript have been improved to a quality of 300 dpi. For Figures 4h, 5h, 6h, 7, 8, and 9, vertical guide lines have been added to facilitate interpretation.
- Conclusions
Add the research's positive points and the potential transformative benefits it can promote. Present some inspiring suggestions for future studies.
Answer: We agree with the Reviewer’s comment. It has been improved.
- References
The article's references do not currently comply with the journal's standards. I suggest the authors to review the standards presented in the submission template and make the necessary corrections. This will ensure the credibility and robustness of your research, giving you and your work the confidence it deserves.
Answer: All literature references have been corrected with Mendeley using the "Multidisciplinary Digital Publishing Institute" style found in the program.
Reviewer 2 Report
Comments and Suggestions for Authors
The introduction needs to be improve by including recent publications (Bacterial Adhesion on Prosthetic and Orthotic Material Surfaces, Coatings, 2021, 11, 1469).
How big is the surface energy?
What about the surface charge. Did authors try to measure it?
Is this material biocompatible?
Author Response
PhD, DSc Anna Masek
Institute of Polymer and Dye Technology
Lodz University of Technology
90-537 Lodz, Stefanowskiego 16, Poland
Tel.: +48 42 631 32 23, Fax: +48 42 636 25 43
May 30, 2024
Dear Editor,
We resubmit our revised paper entitled “Ecologically modified leather of bacterial origin” bywith a request for reconsideration for publication in "Materials” journal.
We have carefully analyzed the comments of the reviewers. The manuscript was corrected exactly according to them. A list of responses to reviewers’ comments and corrections made in the manuscript are attached. In the manuscript, changes made based on the reviewers' comments are marked in red.
The manuscript has not been previously published, is not currently submitted for review to any other journal, and will not be submitted elsewhere before a decision is made by this journal.
For correspondence please use the following information:
Corresponding author: Anna Masek
Institute of Polymer and Dye Technology
Lodz University of Technology
90-537 Lodz, Stefanowskiego 16, Poland
Tel.: +48 42 631 32 93
Fax: +48 42 636 25 43
e-mail: anna.masek@p.lodz.pl
Yours sincerely,
PhD, DSc Anna Masek
Reviewer #2
- The introduction needs to be improve by including recent publications (Bacterial Adhesion on Prosthetic and Orthotic Material Surfaces, Coatings, 2021, 11, 1469).
Answer: We agree with the Reviewer’s comment. It has been improved.
- How big is the surface energy?
Answer: Surface energy measurements have not been made.
- What about the surface charge. Did authors try to measure it?
Answer: Surface charge measurements have not been made.
- Is this material biocompatible?
Answer: Biocompatibility tests of the examined material were not conducted.
Reviewer 3 Report
Comments and Suggestions for Authors
The manuscript details the ecological modification of bacterial nanocellulose through the incorporation of vegetable oils such as rapeseed oil, linseed oil, and grape seed oil as plasticisers. Additionally, it explores the impact of these natural-origin compatibilizers (such as chlorophyll, curcumin, and L-glutamine) on the plasticisation process of BNC, including. This research addresses the conventional challenge of bacterial nanocellulose becoming stiff and brittle when drying, offering a solution by utilisation of plasticised BN for further industry application. While the manuscript’s thoroughness and depth commend it for potential publication in Materials, minor revisions are required for acceptance. Specific suggestions are outlined below:
1. The introduction section lacks sufficient background information and would benefit from the integration of recent articles published within the last three years to enhance the contextual framework.
Author Response
PhD, DSc Anna Masek
Institute of Polymer and Dye Technology
Lodz University of Technology
90-537 Lodz, Stefanowskiego 16, Poland
Tel.: +48 42 631 32 23, Fax: +48 42 636 25 43
May 30, 2024
Dear Editor,
We resubmit our revised paper entitled “Ecologically modified leather of bacterial origin” bywith a request for reconsideration for publication in "Materials” journal.
We have carefully analyzed the comments of the reviewers. The manuscript was corrected exactly according to them. A list of responses to reviewers’ comments and corrections made in the manuscript are attached. In the manuscript, changes made based on the reviewers' comments are marked in red.
The manuscript has not been previously published, is not currently submitted for review to any other journal, and will not be submitted elsewhere before a decision is made by this journal.
For correspondence please use the following information:
Corresponding author: Anna Masek
Institute of Polymer and Dye Technology
Lodz University of Technology
90-537 Lodz, Stefanowskiego 16, Poland
Tel.: +48 42 631 32 93
Fax: +48 42 636 25 43
e-mail: anna.masek@p.lodz.pl
Yours sincerely,
PhD, DSc Anna Masek
Reviewer #3 The manuscript details the ecological modification of bacterial nanocellulose through the incorporation of vegetable oils such as rapeseed oil, linseed oil, and grape seed oil as plasticisers. Additionally, it explores the impact of these natural-origin compatibilizers (such as chlorophyll, curcumin, and L-glutamine) on the plasticisation process of BNC, including. This research addresses the conventional challenge of bacterial nanocellulose becoming stiff and brittle when drying, offering a solution by utilisation of plasticised BN for further industry application. While the manuscript’s thoroughness and depth commend it for potential publication in Materials, minor revisions are required for acceptance. Specific suggestions are outlined below:
- The introduction section lacks sufficient background information and would benefit from the integration of recent articles published within the last three years to enhance the contextual framework.
Answer: We agree with the Reviewer’s comment. The introduction has been enriched with additional content.
Reviewer 4 Report
Comments and Suggestions for Authors
The submitted manuscript is very interesting and brings new knowledge in the production of ecological material based on bacterial nanocellulose. The created methodology provides important and valuable knowledge that shows the way for further research. Nevertheless, I have a few comments.
1) The subject of the study is the investigation of the influence of selected substances on BNC, but the key on the basis of which these substances were selected is not given. It is necessary to better explain this intention of the authors in the text.
2) In the mentioned graphic outputs Fig.1-3, peaks are shown in cases a)-f), i.e. those increased values at the given Wavenumber. It would be necessary to explain it better here.
3) In Fig.3 in case c) on the y axis there is a formal error in the name.
I liked the mentioned research and the increase in the plasticity of this material is certainly very innovative from the point of view of its use. Maybe I missed it, but I don't know if it wouldn't be good to use increased pressure as well. In the production of plastics, linear polyethylene was produced, which showed high fragility, but by using higher pressure and temperature, as well as adequate time, a better connection of macromolecules was achieved. Branched and later cross-linked PEs were formed, which showed better plasticity properties. Of course, this is just an idea for further research.
Author Response
PhD, DSc Anna Masek
Institute of Polymer and Dye Technology
Lodz University of Technology
90-537 Lodz, Stefanowskiego 16, Poland
Tel.: +48 42 631 32 23, Fax: +48 42 636 25 43
May 30, 2024
Dear Editor,
We resubmit our revised paper entitled “Ecologically modified leather of bacterial origin” bywith a request for reconsideration for publication in "Materials” journal.
We have carefully analyzed the comments of the reviewers. The manuscript was corrected exactly according to them. A list of responses to reviewers’ comments and corrections made in the manuscript are attached. In the manuscript, changes made based on the reviewers' comments are marked in red.
The manuscript has not been previously published, is not currently submitted for review to any other journal, and will not be submitted elsewhere before a decision is made by this journal.
For correspondence please use the following information:
Corresponding author: Anna Masek
Institute of Polymer and Dye Technology
Lodz University of Technology
90-537 Lodz, Stefanowskiego 16, Poland
Tel.: +48 42 631 32 93
Fax: +48 42 636 25 43
e-mail: anna.masek@p.lodz.pl
Yours sincerely,
PhD, DSc Anna Masek
Reviewer #4
- The subject of the study is the investigation of the influence of selected substances on BNC, but the key on the basis of which these substances were selected is not given. It is necessary to better explain this intention of the authors in the text.
Answer: We agree with the Reviewer’s comment. An explanation of the criteria for selecting the substances used for modification has been added to the text.
- In the mentioned graphic outputs Fig.1-3, peaks are shown in cases a)-f), i.e. those increased values at the given Wavenumber. It would be necessary to explain it better here.
Answer: We agree with the Reviewer’s comment. Absorbance increases occurring on FT-IR spectra are explained.
- In Fig.3 in case c) on the y axis there is a formal error in the name.
Answer: We agree with the Reviewer’s comment. It has been improved.
Round 2
Reviewer 1 Report
Comments and Suggestions for Authors
The authors met most of the requested adjustments.
Reviewer 2 Report
Comments and Suggestions for Authors
accept